# Automated stance detection in complex topics and small languages: The challenging case of immigration in polarizing news media

Mark Mets[1,2]*, Andres Karjus[1,2,3]*, Indrek Ibrus[2,4], Maximilian Schich[2,4]

1 School of Humanities, Tallinn University, Tallinn, Estonia, 2 ERA Chair for Cultural Data Analytics, Tallinn University, Tallinn, Estonia, 3 Estonian Business School, Tallinn, Estonia, 4 Baltic Film, Media and Arts School, Tallinn University, Tallinn, Estonia

* mark.mets@tlu.ee (MM); andres.karjus@tlu.ee (AK)

**Data Availability Statement:** Data and code used in this study are open access and available in this GitHub repository: https://github.com/markmets/immigration-prediction-EST.

## Abstract

Automated stance detection and related machine learning methods can provide useful insights for media monitoring and academic research. Many of these approaches require annotated training datasets, which limits their applicability for languages where these may not be readily available. This paper explores the applicability of large language models for automated stance detection in a challenging scenario, involving a morphologically complex, lower-resource language, and a socio-culturally complex topic, immigration. If the approach works in this case, it can be expected to perform as well or better in less demanding scenarios. We annotate a large set of pro- and anti-immigration examples to train and compare the performance of multiple language models. We also probe the usability of GPT-3.5 (that powers ChatGPT) as an instructable zero-shot classifier for the same task. The supervised models achieve acceptable performance, but GPT-3.5 yields similar accuracy. As the latter does not require tuning with annotated data, it constitutes a potentially simpler and cheaper alternative for text classification tasks, including in lower-resource languages. We further use the best-performing supervised model to investigate diachronic trends over seven years in two corpora of Estonian mainstream and right-wing populist news sources, demonstrating the applicability of automated stance detection for news analytics and media monitoring settings even in lower-resource scenarios, and discuss correspondences between stance changes and real-world events.

## Introduction

Understanding complex socio-political and cultural issues, such as polarization and news biases requires a comprehensive perception of cultural systems. Computational and data-driven research can offer valuable insights, provided that we acknowledge the limitations of computational methods and are able to scrutinize the findings. Advances in natural language processing, such as pretrained large language models (LLMs) have enabled the analysis of large volumes of data, but these methods may have limited applicability in smaller languages with limited training data and NLP resources. This includes dealing with politically charged issues

**Funding:** M.M., A.K., I.I., and M.S. are supported by the CUDAN ERA Chair project for Cultural Data Analytics at Tallinn University, funded through the European Union Horizon 2020 research and innovation program (Project No. 810961). The funder had no role in study design, data collection and analysis, decision to publish, or preparation of the manuscript. M.M., A.K. and I.I. received funding from Estonian media publishing company AS Ekspress Grupp (https://www.egrupp.ee/en/). The funder provided part of the data but had no role in study design and analysis, decision to publish, or preparation of the manuscript. There was no additional external funding received for this study.

**Competing interests:** The authors have declared that no competing interests exist.

that involve diverse linguistic expressions and cultural perspectives. However, quantifying the reporting of different arguments or stances towards various issues can help scholars to better understand media ecosystems, study the political positions of different media groups or specific outlets, but can also aid industry, including media organizations if they are looking to balance and avoid bias in their reporting.

We report on an experiment of automatically classifying topic-specific political stance in news media texts written in a low to medium-resource, morphologically complex language, Estonian, spoken natively by about 1.1 million people, primarily in the European state of Estonia. While we use one small language as the example, we argue below that our results have implications for the applicability of automated stance detection and media monitoring more broadly. The topic in question is the globally much disputed and often polarizing topic of immigration. Our corpus consists of news articles published in 2015–2022 by one mainstream media group (Ekspress Grupp), and one right-wing populist online news and opinion portal (Uued Uudised, or the "new news"). The study is based on an academia-industry collaboration project with the Ekspress media group, who provided data from their in-house publishing database (but did not influence the design of the study nor the conclusions). Their interest was to assess the neutrality of their content. The goal of this study was to determine the feasibility and accuracy of automated stance detection for linguistically and culturally complex issues (in this case, immigration), in a lower-resource language, and also apply it to mapping stance in a large corpus of news dealing with the topic. This could be applied to assess the balance of different views in news reporting as well as to foster discussions about bothsidesism. For that purpose we compare sources that may likely have contrasting views on the politically charged topic of immigration. We focus on testing a supervised learning approach, annotating a set of training data, tuning a number of different LLMs on the training examples, and testing them on a holdout test set. The best-performing model is further applied to the larger corpus to estimate the balance of different stances towards immigration in the news.

Our experiment design follows a fairly standard annotate-train-test procedure. We first extracted 8000 sentences from the joint corpus using a lexicon of topic-relevant keywords and word stems (referring to keywords such as *migrant*, *immigration*, *asylum seeker*), carried out manual stance annotation, and fine-tuned a number of pre-trained LLMs on this dataset for text classification, including multilingual and Estonian-specific ones. We also experiment with zero-shot classification in the form of instructing GPT-3.5 (note that when this research was being carried out, OpenAI's newer GPT-4 model was not yet available) to classify sentences according to similar guidelines as the human annotators. All LLM classifiers achieve reasonably good test set accuracy, including the zero-shot variant, which performs almost on par with the best annotations-tuned model. Our work has three main contributions.

We demonstrate the feasibility and example accuracy of what amounts to a proof of concept for an automated political stance media monitoring engine, and also compare it to cheaper approaches of bootstrapping a general sentiment analysis classifier to estimate stance, and using zero-shot learning. While not perfect, we argue that the approach can yield useful results if approached critically, keeping the error rates in mind. We have chosen a socio-politically complex example topic and a lower-resource language for this exercise. Consequently, it is reasonable to expect higher accuracy when following analogous procedures, where conditions are more favorable: either the target language having larger pre-trained models available, the topic being of lesser complexity, or larger quantities of training data are annotated. We are making our annotated dataset of 7345 sentences public, which we foresee could be of interest to the Estonian NLP as well as media and communications studies communities, as well as applications of multilingual NLP and cross lingual transfer learning. We also contrast the more traditional annotations based training approach to zero-shot classification. We offer a perspective

of such an approach's future importance in academia and beyond. While first attempts at benchmarking the new generation instructable GPTs as zero-shot classifiers have mostly focused on large languages like English, we provide insight into its performance on a lower-resource language. The topic is also an example of real-world commercial interest, where our industry partner has been interested in keeping balance of their reporting of different stances.

Secondly, we carry out qualitative analysis of the annotation procedure and model results, highlighting and discussing difficulties for both the human annotators and the classifier, when it comes to complex political opinion, dog-whistles, sarcasm and other types of expression requiring contextual and cultural background knowledge to interpret. Lessons learned here can be used to improve future annotation procedures.

Finally, we show how the approach could be used in practice by media and communications scholars or analytics teams at news organizations, by applying the trained model to the rest of the corpus to estimate stances towards immigration and their balance in the two news sources over a 7 year period. This contributes to understanding immigration discourse, media polarization and radical-right leaning media on the example of Estonia. We find and discuss qualitative correspondences between changes in stance and relate them to events such as Estonian parliamentary elections in 2019 and the start of the Russian invasion to Ukraine of 2022.

We therefore aim to fill multiple gaps in existing research: applications of LLMs both in supervised and zero-shot learning contexts to lower-resource languages, better understanding of far-right populism, media polarization and immigration, and work towards automated trend analysis of these topics, in particular but not limited to the Estonian context.

## Analytic approach

We approach stance detection as determining favorability toward a given (pre-chosen) target of interest [1] through computational means. Stance detection (or stance classification, identification or prediction) is a large field of study, partially overlapping with opinion mining, sentiment analysis, aspect-based sentiment analysis, debate-side classification and debate stance classification, emotion recognition, perspective identification, sarcasm/irony detection, controversy detection, argument mining, and biased language detection [2, 3]. Stance detection is used in natural language processing, social sciences and beyond in order to understand subjectivity and affectivity in the forms of opinions, evaluations, emotions and speculations [4]. Compared to the more general sentiment analysis, stance detection is a more topic-dependent task that requires a specific target [1] or a set of targets [5, 6], as does aspect-based sentiment analysis, which is commonly applied to product reviews [7]. We seek to assess stance towards one target, immigration, and contrast our results with a more general sentiment analysis classifier.

Both sentiment analysis and stance detection are classification tasks with multiple possible implementations. Earlier approaches were based on dictionaries of e.g. positive and negative words, and texts would be classified by simply counting the words, using rules of categorization, or various statistical models. We employ the method of tuning large pretrained language models like BERT [8] as supervised text classifiers. Such context-sensitive language models have been shown to work well across various NLP tasks and typically outperform earlier methods [8, 9]. Reports on using LLMs for stance detection in lower-resource languages are relatively limited in literature. However, their value becomes evident in situations where language-specific NLP tools and resources, like labeled training sets, may be scarce, but where there exists ample unlabeled data, such as free-running text, to train a LLM or incorporate the language into a multilingual model [10, 11]. Resources pertinent for NLP encompass both available methods as well as datasets among other factors (cf. [12]).

Automated stance detection has also been applied in studies on immigration and related topics. The data they use is usually textual, ranging from often studied Twitter [2, 13] to online discussion forums [14] and comments of online news [15]. In the context of news media, the immigration topic is also relevant in hate-speech detection, which applies similar methods [13]. These studies use a variety of methods for stance detection, including LLMs. Including single-shot studies (e.g. [16]) where training set topics match the predicted topics; multi-shot approaches which offer partial transferability; and zero shot [6, 15] which aims to predict topics not contained in the training set. Automated stance detection has been used to study immigration topics in less-resourced languages, like Swedish [14], and across-topics (zero-shot) and multilingual approaches using LLMs have been shown to work across languages other than English, like in Italian, French and German [6].

## Object of analysis

Immigration has witnessed increased focus in media and politics in Europe since the 2015 European migrant crisis, but is also relevant globally. Analysis of media representations of immigration is crucial, as it can determine stances towards immigration [17, 18], such as perception of the actual magnitude of immigration. In turn, exposure to immigration related news can have an impact on voting patterns [17]. This topic is also central in populist radical right rhetorics [19, 20]. Social media has been argued to be one of the means for achieving populistic goals [21]. In the Estonian context, most of the radical right content circulating in Estonian-language social media have been reported to be references to articles from the news and opinion portal Uued Uudised [22], making it a relevant source for understanding radical right populists' perspective towards immigration.

Focus on immigration fits into populistic rhetoric in the context of distancing the "us" from the strange or the "other". In the case of the radical right, this other may be often chosen based on race or ethnicity [23]. Such exclusionism of immigrants and ethnic minorities can be present in radical right populism to the extent that it becomes its central feature [19, 20]. Targeted minority groups depend on cultural context and may change over time. For example, before 2015, Central and Eastern European (CEE) populist radical right parties used to target mainly national minorities, whilst in Western-Europe it was more often immigrants. After the 2015 immigration crisis, immigrants also became the main target in the CEE countries [22].

The same applies to Estonia, where immigration has been one of the topics used by the radical right parties to grow their political impact, especially since 2015 [22]. 2015 also marked the emergence of many anti-immigrant social media groups, blogs and online news and opinion portals which have gained popularity since then. This includes the radical right online news portal Uued Uudised, a channel whose news are often ideologically in line with and give voice to the political party of EKRE ("Conservative People's Party of Estonia"). In academic literature EKRE has often been classified as a radical right populist party [22, 24–27] whilst the party describes itself as national conservative [26, 28]. Uued Uudised has been described as both alternative [22] as well as hyper partisan media [28]. It was established in 2015 during the EU immigration crisis. The content of the Estonian radical right media discourse is often following provocative and controversial argumentations [22]. Immigrants are often constructed as an antithetical enemy, where the Other is portrayed as a mirror image of the Self, whereas the Other may first be given negative characteristics that are then perceived as nonexistent in one's own group [22]; cf. [29, 30]. Such othering towards immigration can also be noticed through the topics discussed in the media more broadly, such as framing immigration in the context of criminal activity [25, 31].

While our study has a methodological focus, we have chosen an example that also contributes to a better understanding of the topic of immigration, media polarization and radical-right discourse in our example country of Estonia. Radical-right discourse has been an under-researched topic in Estonian context [22]. Political science has focused on communication of the parties themselves [27, 28, 32], while textual analyses have often focused on social media [22, 26, 30]. These qualitative studies can benefit from a large-scale quantitative approach through automated stance and sentiment detection offering a complementary perspective.

## Methods and materials

### Dataset

We chose the data based on accessibility, and to contrast two sources expected to have different stances on immigration. The corpus consists of articles from 2015 to the beginning of April 2022. The mainstream news are from the Ekspress Grupp, one of the largest media groups in the Baltics. Our data covers one dominant online news platform, Delfi, across all of the time period, and a sample from multiple other daily and weekly newspapers and smaller magazines. The populist radical-right leaning media is represented by the abovementioned online news portal Uued Uudised.

We acquired the Ekspress Grupp data directly from the group and scraped Uued Uudised from its web portal. Both datasets were cleaned of tags and non-text elements. We included Estonian language content only (the official language of the country is Estonian, but there is a sizable Russian speaking minority, and both news sources include Russian language sections). Our dataset consists of 21 667 articles from Uued Uudised (April 2015 to April 2022) and 244 961 from Ekspress Group (January 2015 to March 2022). The received data of Ekspress Group was incomplete with a gap in October-December 2019. The data from 2020 onwards contains multiple times more content from other periodicals besides Delfi (see S1 File for detailed distribution of Ekspress data).

We chose sentence as the unit of analysis, instead of e.g. paragraph or article, for three reasons. The length of articles varies greatly, as does the length of a paragraph across articles, and some articles lack paragraph splits. Secondly, longer text sequences may include multiple stances, which may confuse both human annotators and machine classifiers. Thirdly, the computational model, BERT, has an optimal input length limit below the length of many longer paragraphs. It was hoped a sentence would be a small enough unit to represent a single stance on average, but enough context to inform the model. Admittedly, sentence level analysis does have the limitation of missing potentially important contextual information across sentences, as we further discuss in our annotation and classification-error analysis. It is often hard to deduce an opinion from a single sentence length text alone (cf. [1]), but we do expect sentence to be a suitable unit of analysis to indicate changes in rhetoric and large-scale changes across time.

We extracted immigration-related sentences using a dictionary of keywords to cover different aspects related to immigration, implemented as regular expressions (also to account for the morphological complexity of Estonian and match all possible case forms). Previous research on immigration has approached sampling by choosing topic-specific datasets, like immigration related discussion forums [14] or using dictionary based approaches, like Card et al. [16]. We found using predefined keywords as simple and efficient enough for our task. Using text embeddings can provide a good alternative if keywords are harder to limit or have many synonyms [33]. We created a list of keywords sorted into groups representing various aspects of the migration as well as other closely related topics—migration, refugees, foreign workers, foreign students, non-citizens, race, nationality, and terms related to radical-right and liberal opposition (e.g. "multiculturalism") terms (see S1 File for the full list of keywords). This plurality of

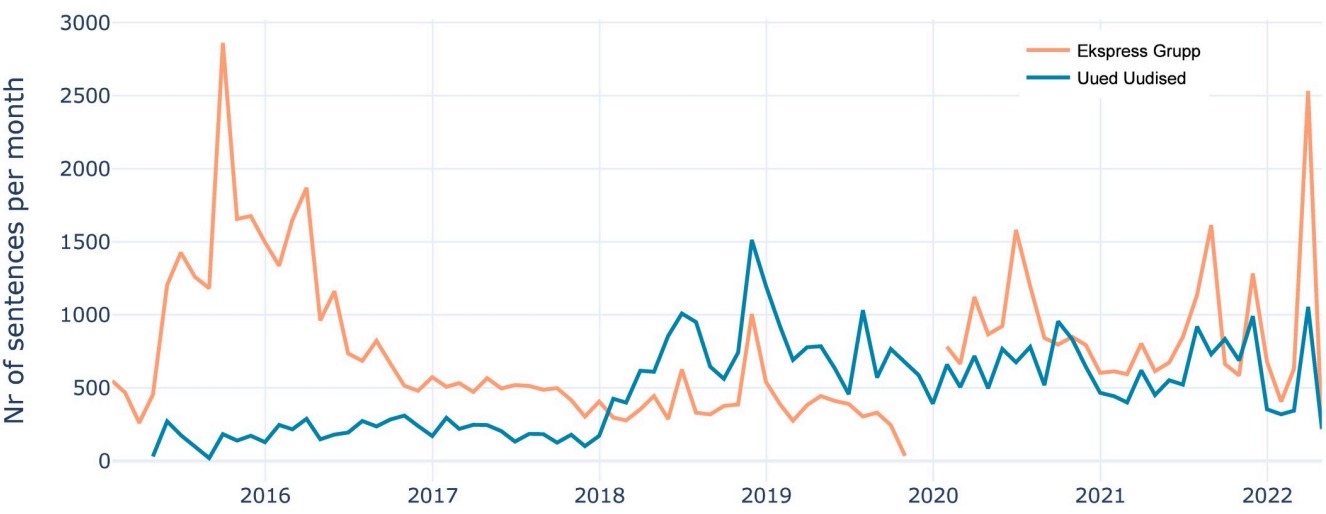

**Fig 1. Monthly distribution of immigration related sentences.** The red line represents the Ekspress Grupp and blue Uued Uudised. There is no data for Ekspress Grupp at the end of 2019, where the count is 0. The change of relevant sentences in Ekspress Grupp after that reflects the difference in the dataset, which was larger and was more varied in terms of specific periodicals (cf. S1 File for distribution of articles per Ekspress Grupp periodical and similar distribution of immigration related sentences but per week).

topics (e.g. also covering "digital nomads") made the task much more challenging but at the same time allowed to grasp more nuances of the migration discourse at large. This yielded sentences that included both opinions as well as factual descriptions and were therefore stylistically varied. In addition to searching for relevant keywords, we used a negative filter to exclude unrelated topics, like bird migration (see Fig 1 for distribution of filtered sentences).

## Annotations

We assigned two Estonian speaking graduate students to annotate a total of 8000 sentences for supervised training. The annotators were compensated monetarily. The sample was balanced by keyword prevalence and publishers (Uued Uudised and Ekspress Grupp), but not by the time or source article. Based on annotator feedback, we removed very long, repetitive or list-like and non-topical sentences, leaving 7345 sentences. The sentences were annotated on a 1–5 point scale from pro- to anti-immigration, with the option to mark the sentence as ambiguous instead. Ratings were later reduced to four classes of Against (1–2), Neutral (3), Supportive (4–5), and Ambiguous. Annotators were instructed to classify sentences expressing supportive or positive attitude as Supportive and the opposite as Against. Neutral class was for the sentences that didn't express either, but were still topical. Ambiguous class was for sentences that were unintelligible, non-topical or expressed multiple stances at once. These were considered unhelpful for model tuning and excluded. Some of the wide variety of sentences were straightforward to interpret while others posed a challenge due to complex metaphorical usage or references requiring additional knowledge. Below are some examples from each classification category, translated into English (for more information and original Estonian see S1 File).

1. "Mass immigration would be disastrous for Europe and it would not solve anything in the world." (Against)

2. "The process to get a residence permit here was not very complicated." (Supportive)

3. "Migration issues must definitely be analyzed, including the aspect of international obligations and their binding nature, and various steps should be considered." (Neutral)

4. "One can only wonder—when do Libyans quit and follow the flow of things when Europe is just talking about controlling the migrant crisis but itself just pours oil on fire." (Ambiguous)

5. "It is not worth mentioning that the person in question is thoroughly Europhile and global-ist." (Against, because the manner presumes that it is said from the perspective of someone who may be against immigration)

6. "They criticize racism, homophobia, xenophobia and what they see as outdated national-ism." (Supportive, but refers to a third person and may thus also be taken as Against)

The sentiment analysis classification used for comparison differed from stance detection only in terms of the annotations used for fine-tuning. We used a publicly available Estonian language dataset of short paragraphs labeled for sentiment as Negative, Neutral, Positive and excluded Mixed class [34].

To calculate inter-annotator agreement, a third annotator (the first author) later annotated a subset from both of the previous annotators which was then compared to each of the original annotators. There was a substantial agreement on Supportive, Again and Neutral classes κ = 0.69 and 0.66 between the third and each of the other annotators (see S1 File for details). There was a very strong agreement if considering only Pro and Against classes (κ = 0.97 for both annotators), indicating that most of the disagreement was between one extreme and Neutral.

## Automatic classifiers

We use our annotated dataset to train and compare several popular LLMs based on BERT and BERT-like [8] transformers architecture: multilingual mBERT [8], XLM-RoBERTa [35], monolingual EstBERT [36] and Est-RoBERTa [37]. We used the larger versions of the publicly available models with 512 tokens to fit longer sentences that were optimal for our setup. We used a Simple Transformers library in Python [38] for working with transformers. To find the best hyperparameters, we minimized the evaluation loss whilst choosing between different batch size, learning rate, epoch and warm up ratio. The best parameters were batch size 16, learning rate 5e-5, 2 epochs and warmup ratio 0.1 for EstBert, and 16, 5e-6, 5e, and 0.1 for XLM-RoBERTa. The chosen learning rate matches what is often recommended for these types of models. Dataset was split 80/20 into training and test sets and results were cross-validated as averages from 5 runs with different sampling. As the training and test data were unbalanced in terms of the number of classes, our training took into account the weights (relative size) of the classes. We used F1 score as the evaluation metric, which is preferable to simple accuracy alone given the unbalanced classes.

In addition, we also compared the results with that of GPT-3.5 (we conducted our experi-ments on March 3, 2023, using the February 13 version of GPT-3.5). The new approach of using (even larger) generative LLMs as zero-shot classifiers (also known as prompt-based learning, cf. [39]) has opened up new avenues of potentially cheap and efficient text classifica-tion, as it requires no fine-tuning and can simply be instructed using natural language. A growing body of research has also shown that this prompt-based learning approach can rival specialized NLP models as well perform on par with human annotators [40–42].

There has been a surge of research on GPT performance for different NLP tasks, but mostly focused on English. It has been shown that GPTs can achieve similar or better results in English than comparable supervised and other zero-shot models, including in stance detection [43, 44]. On a wider array of tasks, GPTs have been shown to be good generalist models, but performing worse than models fine-tuned for a specific task [45]. On the other hand, closed source commercial models like GPT-3.5 are problematic in terms of biases, evaluation and

replicability due to their ongoing development and closed nature [46]. Our goal is to estimate its potential relevance for future studies by comparing it with the established pipeline of supervised tuning of pretrained LLMs for classification tasks. While we make use of this commercial cloud service LLM due to the current lack of options for the Estonian language in terms of GPT-3.5 class models, developments in the open source LLM landscape hold promise for more accessible and replicable applications in the future.

We created a prompt that included optimized classification instructions and input sentences, in batches of 10. Responses not falling into Against, Neutral or Supportive classes were requested again until only labels belonging to this set were returned. Also if a wrong number of tags was returned, the sentences were requested again. An example input and output would look as follows:

**Input**: *Stance detection. Tag the following numbered sentences as being either "supportive", "against" or "neutral" towards the topic of immigration. "Supportive" means: "supports immigration, friendly to foreigners, wants to help refugees and asylum seekers". "Against" means: "against immigration, dislikes foreigners, dislikes refugees and asylum seekers, dislikes people who help immigrants". "Neutral" means: "neutral stance, neutral facts about immigration, neutral reporting about foreigners, refugees, asylum seekers". Don't explain, output only sentence number and stance tag.*

*1. Unfortunately, by now the violence has seeped from immigrant communities to all of the society.*

*2. [. . .]*

**Output**:

*1. Against*

*2. [. . .]*

## Results

The best-performing fine-tuned model was based on Est-RoBERTa, achieving an acceptable F1 macro score of 0.66 (precision 0.65 and recall 0.68; see Table 1). The difference with other monolingual EstBert (0.64) and multilingual XLM-RoBERTa (0.64) was minimal. All of the fine-tuned models performed better at classifying Against than Supportive stances. Est-RoBERTa model achieved F1 0.74 for Against, 0.69 for Neutral and 0.55 for Supportive class. The misclassification was mostly between Neutral and one extreme (see Fig 2), similarly to e.g. Card et al. [16]. We regard it preferable to confusing the two extremes. The results are comparable to similar studies, and there is little difference between the models. Classifier trained on an existing sentiment dataset with Est-RoBERTa achieved the worst score but performed better than expected. There were more mistakes between the two extremes than with sentiment analysis training set (see S1 File for sentiment confusion matrix). We confirmed sentiment analysis training set performance by comparing the sentiment and stance predictions for all of the immigration related sentences, resulting in a fair agreement (kappa 0.29). It demonstrates the complexity of our task, which included features from stance as well as sentiment. Finally, comparable performance of zero-shot GPT-3.5 with the best model shows it could serve as a viable but cheaper alternative to fine-tuned models.

We further assessed the mistakes made by the best performing classifier. We looked at the mistaken predictions in the evaluation set between Against and Support classes and observed at least four types of interpretable mistakes.

1. Mistaken human annotations. These may be hard to fully exclude when using human annotations but could be reduced with better instructions.

2. Sarcasm, a well known challenge in NLP

3. Ambiguous and context dependent sentences. These may be generally more complicated to classify

4. Sentences that refer to a third person. These are tricky, as referencing someone else's opinion may implicitly imply agreeing or opposite standpoint, which is highly context dependent and therefore not easy, but a simpler task for humans than classifiers. These could relate to our chosen unit of analysis; paragraphs might perform better.

**Table 1. Comparison of classification models.** F1 scores from different models by each class and across all classes. Bold indicates the best result with Est-RoBERTa. We used 5-fold cross-validation with 20% of data with all models (see S1 File for more detailed results).

| Model | Against | Neutral | Supportive | F1 macro |
|---|---|---|---|---|
| Naive Bayes | 0.59 | 0.58 | 0.38 | 0.52 |
| EstBert class | 0.69 | 0.70 | 0.53 | 0.64 |
| **Est-RoBERTa** | **0.74** | **0.69** | **0.55** | **0.66** |
| XLM-RoBERTa | 0.73 | 0.65 | 0.54 | 0.64 |
| mBERT (cased) | 0.66 | 0.64 | 0.40 | 0.56 |
| mBERT (uncased) | 0.64 | 0.58 | 0.38 | 0.54 |
| GPT 3.5 | 0.74 | 0.64 | 0.57 | 0.65 |
| Est-RoBERTa Sentiment | 0.63 | 0.42 | 0.42 | 0.49 |

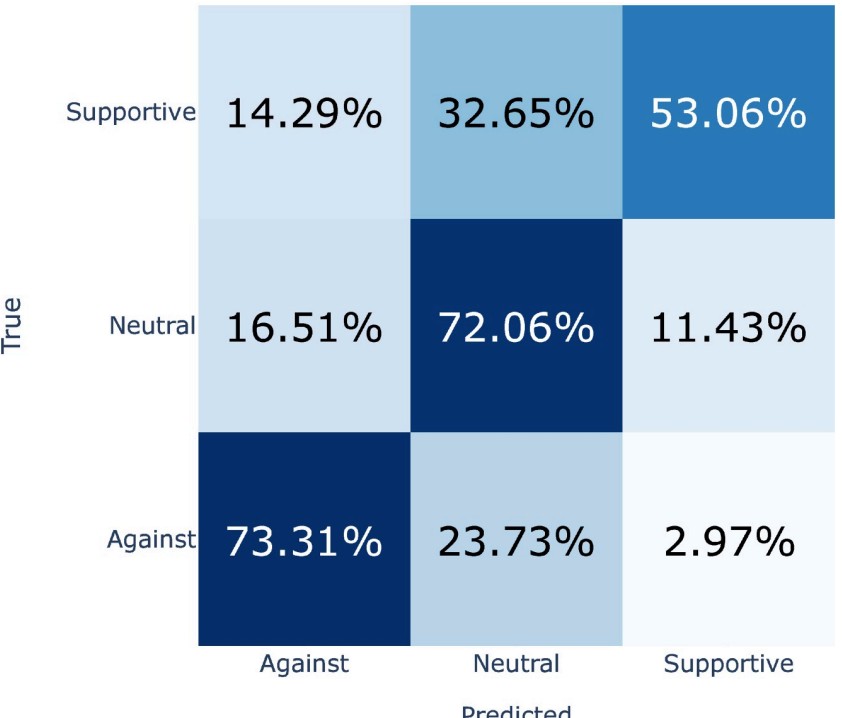

**Fig 2. Confusion matrix of stance detection.** Based on one fold from the best performing model (testing dataset not used for training). Percentage shows the overlap between true (annotated) and predicted classes. Ideal but non-realistic classification would be 100% for diagonal from bottom left to top right. We regard the small values in top left and bottom right as a good sign, showing that most of the mistakes were between Supportive or Against, and Neutral, not between the two extremes (see S1 File for comparison with sentiment analysis).

## Limitations

The limitations of classifier performance are at least partly rooted in human annotations. Some of these shortcomings were reported by the annotators themselves. The distinction between neutral and ambiguous classification was also problematic, where more clear instructions might have helped. Confusion between neutral and ambiguous classes is not expected to have a strong effect on the results, but may have limited the size of our training set by having neutral sentences classified as ambiguous. Annotations are also dependent on the annotators' prior knowledge and biases. Annotators were instructed to only rate the sentence itself, but we expect that they also relied on personal contextual knowledge (cf. [12] for more aspects for annotators to consider in future studies). Yet, LLMs are not impervious to (e.g. training set induced) biases either. It may explain why in some cases smaller and more specific models might perform better [47].

We suspect that some limitations to classifier accuracy arose from the dataset itself. The text contained opinions, descriptive sentences as well as quotations in indirect as well as direct speech. This was discussed with the annotators before and during the annotation process, as it was reported to have created some confusion. In the case of opinions, explicit expressions were easily distinguishable, but in many cases the opinions were implicit. Quotations were also problematic as these could easily be misinterpreted without the proper context that a paragraph might provide. Sarcasm and metaphoric speech is also among challenges that automatic classifiers have to face, e.g. "The protests were but shouts in the deserts because the wheel of racial equality had already been set on its way." (*Kuid protestid jäid hüüdjaks hääleks kõrbes, sest rassilise võrdsuse ratas oli juba hooga veerema lükatud*.). We also included keywords often used by the radical right to negatively refer to the liberals, like "multiculturalists" and "globalists" etc., which may be difficult to interpret as against immigration without context or prior knowledge. Annotators also reported pro immigration stances as harder to identify. This may be due to anti-immigration rhetoric being more systematic and less fragmented whilst pro-immigration rhetoric is more dependent on the specific sub-topic.

## Exemplary analysis

Lastly, we conducted an exemplary diachronic analysis of the change of stances towards immigration across time. This tests the applicability of our method and demonstrates some of its possible uses. In the following, we visualize and analyze the larger changes in the stance trends in relation to media events, look at the related media polarization and general similarities based on text embeddings.

The relative amount of immigration related articles across time and publisher (see Fig 3) provides an understanding of immigration related media events and their importance for each of the publishers. Uued Uudised clearly focuses more on the immigration topic than Ekspress Grupp, based on keyword prevalence. Uued Uudised also has a stronger reaction to immigration related media events, such as the European migration crisis of 2015–2016, UN immigration pact at the end of 2018, and the Russian invasion of Ukraine from February 2022 onwards, which caused an increase in refugees. These findings confirm what is known about radical-right media in general and it provides novel insight into the Estonian context.

We used the best-performing model, based on Est-RoBERTa, to predict the stances of all sentences in the corpus containing relevant keywords (n = 106539). We focus on monthly trends, as a tradeoff between detail and the amount of available data per unit of time.

Trends seen in Fig 4 shows polarization and indicates changes of stance corresponding to the UN migration pact, elections, and Russian invasion of Ukraine. Uued Uudised stance was generally against immigration, not neutral or supportive. On the other hand, Ekspress Grupp

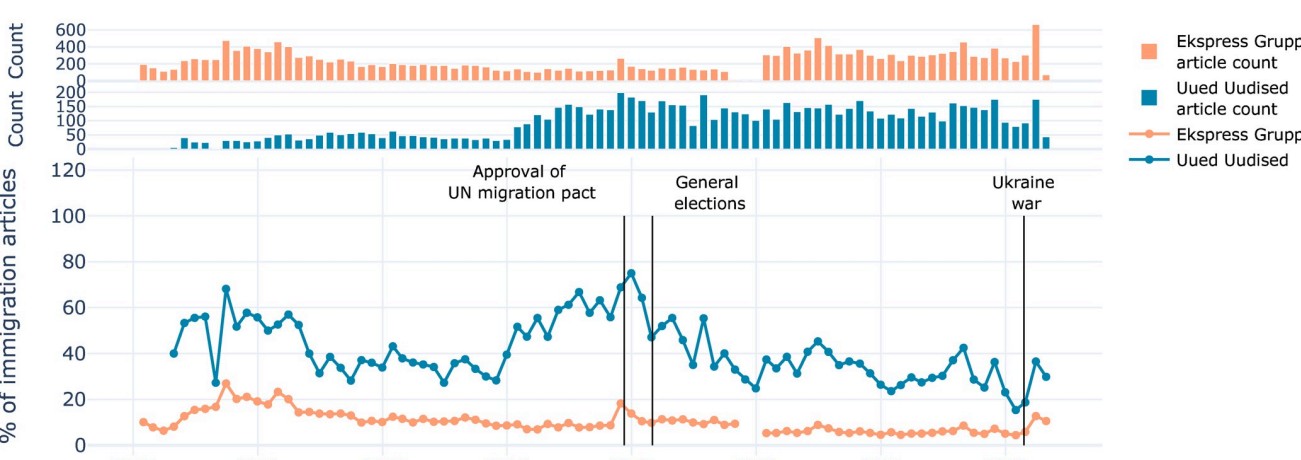

**Fig 3. Percentage of articles mentioning immigration.** Top plots show the counts of articles mentioning immigration. The articles contain at least one immigration related keyword. Higher percentage for the populist radical-right source (blue) confirms that the outlet is more focused on the immigration. The fluctuations in Uued Uudised is due to the smaller amount of data in absolute terms, especially in 2015. The relatively lower amount of immigration related articles in Ekspress group data since 2020 is likely connected to the significantly increased amount of content from a larger variety of specific journals, indicating that the amount of immigration related content is somewhat dependent on specific journals of Ekspress Group (see S1 File for Ekspress Grupp data distribution).

had a dominantly neutral stance over time and kept generally more stable than Uued Uudised. The relative stance differed noticeably per keyword group, whereas multiculturalism and xenophobia and race related words had the highest percentage of sentences labeled as Against migration (see S1 File for stances per keyword groups).

There is a clear change taking place around the 2018–2019 during the UN migration pact discussions (most heated debates in Estonian media happening around November 2018) and general elections (March 3, 2019). Uued Uudised contained more sentences classified as Against migrants than before and right after that period. The share of the Against stance is increasing with the UN migration pact discussions, but decreases soon after the elections in March 2019. The Against stance increased in these years for all of the keyword groups. A change is also noticeable in Ekspress Grupp, where relevance of Against stance increases during the same period. This demonstrates the possible connection between potential politicization of the migration topic and the elections. From March 2020, when Covid-19 became the dominant media event, the stances appear to have shifted again. This may be due to the redirection of focus onto other topics, such as Covid, where the radical-right changed its focus from anti-immigration to anti-governmental. Lastly, the events of the Russian invasion into Ukraine in 2022 correspond to a small increase in supportive stance in Uued Uudised and a much larger increase in supportive stance towards immigrants in Ekspress Grupp. Whilst the Supportive stance increased in almost all of the keyword groups for Ekspress Grupp, there was more variability for Uued Uudised. This ambiguity of Uued Uudised may reflect the continued anti-immigration rhetoric by the related rightwing political party of EKRE.

The findings confirm the relevance of anti-immigration topics in the radical-right populists discourse in general (cf. [19], [20] and their possible growth in Estonia since 2015 (cf. [22]). We also demonstrate how the anti-immigration discourse may have been boosted by the populist radical-right before the time of the elections (cf. [17]). Next we have shown how the immigration discourse was disrupted by the covid outbreak; and the possible changes in attitudes

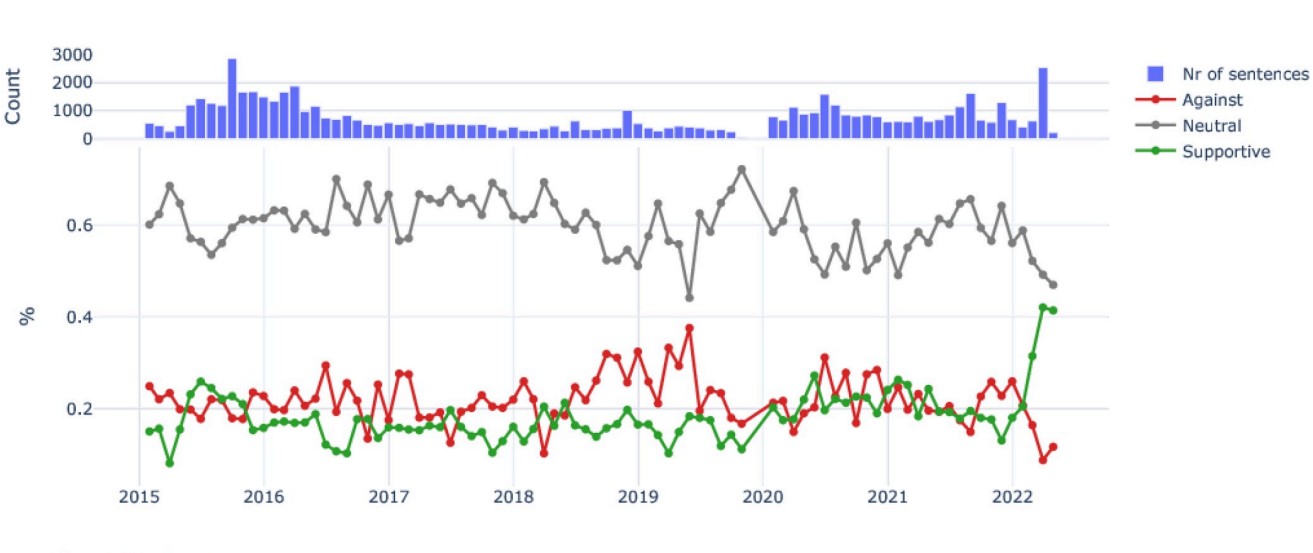

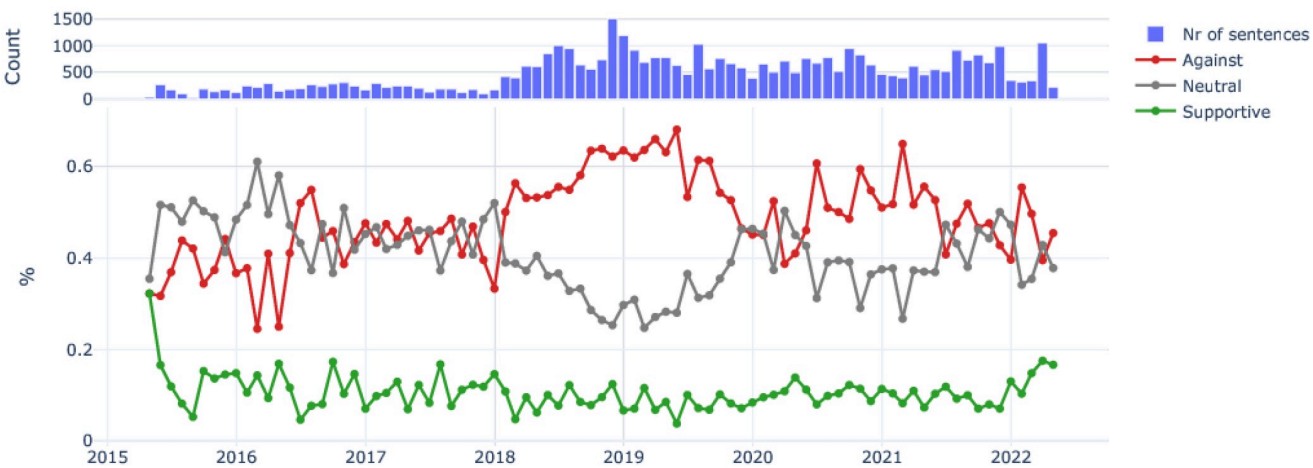

**Fig 4. Stances of immigration related sentences.** Shows the relative percentage of each stance per month for both publishers. Higher value means more sentences classified as that stance in that month. Barplots provide the amount of immigration sentences per month in comparison. In 2022, at the beginning of the Ukraine war, there was a noticeable increase in Supportive stances towards migration in the Ekspress Group with a much smaller increase in Uued Uudised.

towards immigration in the new context of Russian invasion of Ukraine. Most importantly, we provided novel large-scale trends on that specific phenomena of study that calls for further analysis.

## Discussion

Our study shows that automated stance detection is feasible and provides useful insights for media monitoring and analytics purposes, also beyond large languages like English or German. The accuracy of the classifiers was satisfactory, achieving F1 macro 0.66 with Est-RoBERTa. GPT-3.5 achieved a similar result of 0.65. We expect zero-shot accuracy will improve as

generative AI models are being improved and developed. Classification of Against stances was noticeably more accurate than of Supportive stances. As expected, radical-right news media indeed holds generally a more anti-immigration stance in comparison to more mainstream news. We also provided insights into stance change over time, relating it to known local and world events, identifying increased interest towards these topics during the 2015–2016 immigration crisis, the 2018–2019 UN immigration pact and local elections, and the 2022 Russian invasion into Ukraine. These findings, approximated by applying an automated classifier, can be used as basis for further more in-depth research in Estonian-specific or areal media and politics studies.

However, there are also limitations. Fine-tuning pretrained LLMs as classifiers requires annotated training data, which may not be available for specific topics or in lower-resource languages. We discussed issues with annotation, pointing out that linguistically and sociopolitically complex topics such as this are also difficult for human annotators and for formalizing the task. There is also the question of unit of analysis: shorter units like sentences are fast to annotate, but may not contain enough contextual information. Longer like paragraphs do, but may contain multiple stances, which complicates the task both for humans and machines.

## Future research

Although supervised stance detection can provide acceptable results, the need for annotated training data makes it time consuming and expensive, while being applicable to one topic at a time. One option is to use a generic sentiment classifier instead. However, we showed that this does not work very well for complex topics such as immigration, where support may be expressed in sentences with negative overall tone, and vice versa. Using new generation generative LLMs may provide a solution, being easy to instruct in natural language, and applicable across languages, tasks and topics. This makes it particularly attractive for smaller languages with less resources and with less existing annotated datasets.

These models could also be used to annotate data in tandem with human annotators, or augment existing annotations [40]. Accuracy and model bias should still be evaluated. For example, in our case it could have been used to further classify sentences as expressing opinions, factual descriptions, and direct quotes. This can result in a feedback loop that results in better datasets, more accurate models and also better understanding of the functioning of the model through assessing the classification errors.

This new approach has already been explored in preliminary experiments, which our research complements. Zero-shot classifications should still be evaluated just like any other machine learning task, and not be taken for granted. This can be done with annotated datasets, like we have done here, but creating small test sets is generally an easier task compared to large training sets. In summary, human annotations and the development of good practices to carry them out will still be useful and necessary. This is more so the case in smaller languages, where possibly less training data has been used to train multilingual LLMs like RoBERTa or the GPTs. The problems faced are to an extent similar: annotating new datasets requires instructing human annotators, and using generative models requires careful prompt engineering. This also complicates the replication of results, as slightly different instructions can lead to differences in classification performance (in addition to the inherently stochastic nature of generative AI; [48]). The same however holds for human annotators. While replicability and transparency challenges need to be solved, the ready availability of deep learning in the form of cloud services for non-computer scientists and researchers with limited access to large computer clusters, holds great potential, as does the concurrent growing availability of open-source LLMs. Therefore, we expect generative AI driven analytics to become more widespread across

disciplines. This also calls for more critical studies as well as thorough analysis of the applications of the methods to better understand the biases related to specific LLMs and cloud-based services.

## Conclusions

We demonstrated the applicability of automated stance detection using pretrained LLMs for socio-politically complex topics in smaller languages on the example of Estonian news media coverage of immigration discourse. We compare several popular models, and also release the stance-annotated dataset. Our experiments with using an instructable zero-shot classifier are promising, and if applied carefully, this approach could obviate the need for large-scale topic-specific annotation and expedite media analytics and monitoring tasks. This is more so the case in languages where such resources are limited. As a proof of concept, we also applied one classifier to the larger corpus to provide an overview of changes in immigration in Estonian news media in 2015–2022, including one mainstream and one radical-right news source, finding support for discussions in previous literature as well as providing new insights.

## Supporting information

**S1 File.**
(PDF)

## Author Contributions

**Conceptualization:** Mark Mets, Andres Karjus, Indrek Ibrus, Maximilian Schich.

**Data curation:** Mark Mets.

**Formal analysis:** Mark Mets.

**Funding acquisition:** Indrek Ibrus.

**Investigation:** Mark Mets.

**Methodology:** Mark Mets, Andres Karjus, Indrek Ibrus.

**Software:** Mark Mets.

**Supervision:** Andres Karjus, Maximilian Schich.

**Visualization:** Mark Mets.

**Writing – original draft:** Mark Mets, Andres Karjus.

**Writing – review & editing:** Andres Karjus, Indrek Ibrus, Maximilian Schich.

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
