## [Decision Letter · Decision Letter 0]

18 Oct 2023

PONE-D-23-15697Automated stance detection in complex topics and small languages: the challenging case of immigration in polarizing news mediaPLOS ONE

Dear Professor Mark Mets,

Thank you for submitting your manuscript to PLOS ONE. After careful consideration, we feel that it has merit but does not fully meet PLOS ONE’s publication criteria as it currently stands. Therefore, we invite you to submit a revised version of the manuscript that addresses the points raised during the review process.

We look forward to receiving your revised manuscript.

Kind regards,

Pantea Keikhosrokiani

Academic Editor

PLOS ONE

Journal Requirements:

“M.M., A.K., I.I., and M.S. are supported by the CUDAN ERA Chair project for Cultural Data Analytics at Tallinn University, funded through the European Union Horizon 2020 research and innovation program (Project No. 810961). The funder had no role in study design, data collection and analysis, decision to publish, or preparation of the manuscript. M.M., A.K. and I.I. received funding from Estonian media publishing company AS Ekspress Grupp (https://www.egrupp.ee/en/). The funder provided part of the data but had no role in study design and analysis, decision to publish, or preparation of the manuscript”

4. We notice that your supplementary figures are uploaded with the file type 'Figure'. Please amend the file type to 'Supporting Information'. Please ensure that each Supporting Information file has a legend listed in the manuscript after the references list.

5. We notice that your supplementary tables are included in the manuscript file. Please remove them and upload them with the file type 'Supporting Information'. Please ensure that each Supporting Information file has a legend listed in the manuscript after the references list.

Reviewers' comments:

Reviewer's Responses to Questions

**Comments to the Author**

1. Is the manuscript technically sound, and do the data support the conclusions?

Reviewer #1: Yes

Reviewer #2: Partly

2. Has the statistical analysis been performed appropriately and rigorously? 

Reviewer #1: Yes

Reviewer #2: No

3. Have the authors made all data underlying the findings in their manuscript fully available?

Reviewer #1: Yes

Reviewer #2: Yes

4. Is the manuscript presented in an intelligible fashion and written in standard English?

Reviewer #1: Yes

Reviewer #2: No

5. Review Comments to the Author

Reviewer #1: The manuscript explores the application of large language models (LLMs) for automated stance detection in the context of immigration in polarizing news media, focusing on the Estonian language. The study involves annotating a dataset of pro and anti-immigration examples, comparing the performance of multiple LLMs as supervised learners, and investigating diachronic trends over a seven-year period in two corpora of Estonian mainstream and right-wing populist news sources.

The article covers an important and timely topic, and the research objective is clear. The use of LLMs for automated stance detection in a challenging scenario is relevant. However, there are areas where the manuscript could be improved:

1. Abstract: The abstract should provide a concise summary of the study, including the research objective, methodology, main findings, and implications. It currently lacks specific details about the dataset size, performance metrics, and significant findings.

2. Introduction: The introduction provides a good overview of the research context, but more explicit statements about the research gap and the significance of the study must be included. Please clearly articulate how the study contributes to existing literature and highlight the novelty and potential impact of using LLMs for stance detection in lower-resource languages.

3. Methodology: Provide more details about the annotation procedure, including the guidelines given to the human annotators. Explain how inter-annotator agreement was measured and addressed. Also, authors should describe the fine-tuning process of the LLMs, including the hyperparameters used. Please specify the evaluation metrics employed and discuss their appropriateness for the task.

4. Results: The authors evaluate the performance of different models and analyze the changes in stances over time. But below are some suggestions for improvement:

o Provide more details about the dataset used for training and evaluation. Include information such as the size of the dataset, data sources, and any preprocessing steps applied.

o When discussing the performance of different models, you can consider including statistical significance tests to determine if the observed differences are statistically significant.

o Perhaps you can explain the practical implications of achieving an F1 macro score of 0.66. How does it compare to existing methods or benchmarks in the field? Are there specific applications where this level of accuracy is considered acceptable?

5. Limitations:

o Please clarify the limitations attributed to human annotations. What specific challenges were encountered with human annotators? How can these limitations be addressed or minimized in future studies?

o Discuss the potential biases in the dataset and how they might impact the classifier's performance. You may consider exploring methods to mitigate biases or evaluate the impact of biases on the results.

o Please provide more details on the annotation instructions given to the human annotators. How were they trained? Did they receive any feedback or clarification during the annotation process? Clear instructions can help improve the quality of annotations.

6. Exemplary Analysis:

1. I think it is necessary to give a brief overview of the diachronic analysis methodology used for analyzing changes in stances over time. We need to understand the approach taken and the significance of the findings.

2. When discussing the exemplary analysis, more specific interpretations of the findings are needed. For example, what are the implications of the observed trends in stance changes? How do these trends align with existing literature or theories on media polarization and immigration discourse?

7. Future Research:

1. Elaborate on the suggestion of using generative models like ChatGPT for annotating data or augmenting existing annotations. How would this approach work in practice? What are the potential benefits and challenges?

2. Discuss the potential limitations or risks of relying solely on generative models for annotation. Are there any concerns related to biases or the reliability of annotations produced by these models?

3. Provide more details on the proposed evaluation of zero-shot learning and the necessity of annotated datasets. How would the accuracy of zero-shot learning be evaluated? In what scenarios would annotated datasets still be necessary?

Reviewer #2: 1. In the introduction section, mention the key contributions of this text/speech classification point by point.

2. There are a number of classifiers, why did the author choose this particular classifier for this classification problem?

3. Classification accuracy is not too high, how this model can be effective in practice?

4. There are some typos and grammatical anomalies in various sections in this article, overall language quality shall be improved.

6. PLOS authors have the option to publish the peer review history of their article (what does this mean?). If published, this will include your full peer review and any attached files.

Reviewer #1: No

Reviewer #2: No

---

## [Author Response · Author response to Decision Letter 0]

29 Nov 2023

1. In response to the editor's request to correct the funding proposal. We added the sentence "There was no additional external funding received for this study." Here's the corrected version:

M.M., A.K., I.I., and M.S. are supported by the CUDAN ERA Chair project for Cultural Data Analytics at Tallinn University, funded through the European Union Horizon 2020 research and innovation program (Project No. 810961). The funder had no role in study design, data collection and analysis, decision to publish, or preparation of the manuscript. M.M., A.K. and I.I. received funding from Estonian media publishing company AS Ekspress Grupp (https://www.egrupp.ee/en/). The funder provided part of the data but had no role in study design and analysis, decision to publish, or preparation of the manuscript. There was no additional external funding received for this study.

2. We also mention that we added one more affiliation to one of the authors (Andres Karjus - Estonian Business School, Tallinn, Estonia)

---

## [Decision Letter · Decision Letter 1]

3 Apr 2024

Automated stance detection in complex topics and small languages: the challenging case of immigration in polarizing news media

PONE-D-23-15697R1

Dear Dr. Mets,

We’re pleased to inform you that your manuscript has been judged scientifically suitable for publication and will be formally accepted for publication once it meets all outstanding technical requirements.

Kind regards,

Pantea Keikhosrokiani

Academic Editor

PLOS ONE